# Safety and efficacy of 3- and 5-day regimens of levamisole in loiasis: a randomized, placebo-controlled, double-blind clinical trial

Cédric B. Chesnais[1,2] ✉, Marlhand C. Hemilembolo[2], Bachiratou A. Sahm[1], Florentin Toutin[1], Ericson Djeutassong[3], Nadia Nga-Elomo[1], Benjamin Cuer [1], Mas A. Ntsiba-N'Goulou[2], Miveck Pakat[2], Sébastien D. S. Pion[1], François Missamou[2], Michel Boussinesq[1] & Jérémy T. Campillo [1] ✉

Individuals with high *Loa loa* microfilarial densities (MFD) risk serious adverse events (SAEs) following ivermectin treatment. A single dose of levamisole (LEV) induces a temporary, progressive MFD decrease. This double-blind, randomized, placebo-controlled trial evaluated the safety and efficacy of 3-day and 5-day LEV regimens for the treatment of loiasis in the Republic of the Congo. Participants were randomly assigned to receive placebo (PLA), LEV-3, or LEV-5. Safety was assessed by the occurrence of SAEs and the frequency and severity of AEs. Efficacy was measured by changes in *Loa loa* MFD. No SAEs were reported, and AE severity was comparable across treatment arms. On day 5, MFD reductions were greater in the LEV-3 and LEV-5 groups compared to PLA ( −1.6%, 29.3%, and 51.4%, respectively; *P* < 0.001). By day 7, a higher proportion of participants achieved ≥40% MFD reduction in the LEV-5 group (44.4%) compared to LEV-3 (34.6%) and PLA (8.3%) (*P* = 0.051). Post hoc contrasts confirmed significantly greater MFD reduction with LEV-5 versus LEV-3. These findings suggest that a 5-day LEV regimen is safe and moderately effective for reducing *L. loa* MFD and may represent a promising alternative for individualised management in loiasis-endemic areas, particularly where onchocerciasis is hypoendemic and coendemic with loiasis. Trial registration: NCT06252961.

Loiasis is a parasitic infection caused by the filarial nematode *Loa loa*, affecting more than 15 million people in Central Africa[1]. Since the 1990s, loiasis has been a significant challenge to the elimination of onchocerciasis. Indeed, onchocerciasis control relies on mass ivermectin (IVM) treatment of the populations living in meso- and

hyperendemic areas, where the prevalence of onchocercal nodules exceeds 20% in males aged >20 years. However, in Central Africa, the co-endemicity with loiasis has led to serious adverse events (SAEs), including fatal encephalopathy, following IVM treatment in individuals with high densities of *Loa* microfilariae (mf) in their blood[2]. These SAEs

[1]TransVIHMI, Université de Montpellier, INSERM Unité 1175, Institut de Recherche pour le Développement (IRD), Montpellier, France. [2]Programme National de Lutte contre l'Onchocercose, Direction de l'Épidémiologie et de la Lutte contre la Maladie, Ministère de la Santé et de la Population, Brazzaville, Republic of the Congo. [3]Higher Institute for Scientific and Medical Research (ISM), Yaoundé, Cameroon. ✉e-mail: cedric.chesnais@ird.fr; jeremy.campillo@ird.fr

likely result from the rapid paralysis of large numbers of *Loa* mf by IVM, leading to their passive circulation in the bloodstream and subsequent embolization in brain capillaries.

The WHO aims to achieve several milestones by 2030, including verified interruption of onchocerciasis transmission in 12 countries and cessation of mass drug administration (MDA) with IVM in at least one focus area in 34 countries[3]. However, these milestones are out of reach in areas where loiasis and onchocerciasis are co-endemic. To address this challenge, alternative treatment strategies to mass drug administration of IVM have been proposed[4]. One potential approach consists in pre-treating the entire population, or individuals at risk of SAE, with a drug that progressively reduces *Loa* microfilarial density (MFD) below the threshold that is associated with post-IVM neurological SAEs. Several drugs and regimens, including albendazole[5–8], antimalarials[9], and low doses of IVM[10,11], have been tested, however none have proven suitable due to excessive or insufficient effects or high inter-individual variability. A recent trial also suggests that a low dose (2 mg) of moxidectin induces a slower decrease in *L. loa* MFD than a standard dose of IVM[12], but its safety in heavily infected individuals has not yet been evaluated.

To explore a treatment that could gradually reduce *Loa* MFD, enabling its use both as a systematic pre-treatment of the population before implementing IVM MDA to control onchocerciasis, and as an individual treatment for subjects with very high *L. loa* MFD, we previously conducted a clinical trial using a single dose (2.5 mg/kg) of levamisole (LEV) for loiasis[13]. LEV is included in the WHO's List of Essential Medicines and is commonly used at doses of 150 mg (or 2.5 mg/kg) to treat soil-transmitted helminths infection[14]. It has also already shown moderate activity against other filarial species[15–23]. Our first trial showed that a single dose of LEV has a transient and progressive effect on *L. loa* MFD in some subjects, but not in others, making is insufficient for the intended purpose. Therefore, we have now evaluated the safety and efficacy of a standard dose of LEV given daily for 3 or 5 days in individuals infected with *L. loa*.

## Results

### Screening of eligible participants
A total of 2808 individuals were screened in July 2023, of whom 152 met the eligibility criteria for the pre-inclusion phase in June 2024 (Fig. 1). At the inclusion phase, 39 individuals had relocated during the interim period, 2 declined to participate, and 14 were excluded based on non-inclusion criteria. Ultimately, a total of 97 individuals were randomly assigned to one of the three intervention arms.

### Baseline characteristics
Following the application of inclusion and exclusion criteria, 92 participants remained (5 were absent at D1 and did not receive the treatment they were assigned to) (Fig. 1). These 92 participants were included in the safety analysis (PLA = 30, LEV-3 = 33, and LEV-5 = 29). One participant was found to have a *L. loa* MFD of 0 mf/mL prior to treatment. For this reason, and because percentage reduction calculations were not feasible, the modified intention-to-treat (mITT) analysis was conducted on only 91 participants (PLA = 30, LEV-3 = 33, and LEV-5 = 28). Additionally, technical issues related to the slide staining process occurred for 23 individuals at the D7 time point. As the MFD values for this group were deemed unreliable, these individuals were excluded from the per protocol (PP) analyses, leaving the following numbers in the three arms: PLA = 24, LEV-3 = 26, and LEV-5 = 18 (Total: 68). The baseline characteristics of participants are shown in Table 1.

For the mITT efficacy analysis, the mean ages were 50.1 ± 11.2, 46.6 ± 13.1, and 46.4 ± 13.1 years for the PLA, LEV-3, and LEV-5 arms, respectively. The proportions of male participants were 66.7%, 69.7%, and 64.3% in these same arms, respectively. For the PP efficacy analysis, the mean ages ± SD were 50.5 ± 11.6 years, 47.5 ± 13.6 years, and 45.3 ± 12.9 years for the PLA, LEV-3, and LEV-5 arms, respectively. The

proportions of male participants were 66.7%, 69.2%, and 66.7%, respectively.

In the mITT efficacy analysis, the medians and IQR of *L. loa* MFD are reported in Table 1 (same sample size) for the PLA and LEV-3 arms, and were 11,817 (6882–24,635), and 400–130,230 mf/mL for the LEV-5 arm. In the PP efficacy analysis, medians and IQR of these same variables were 16,332 (8170–27,522) mf/mL in the PLA arm, 13,120 (8025–24,675) in the LEV-3 arm, and 16,277 (5650–27,580) mf/mL in the LEV-5 arm.

### Safety assessment
A total of 75 adverse events (AE) were reported during the trial. Of these, six were considered unrelated to the intervention, based on chronologic or clinical criteria: one long-standing pruritus, one case of lumbago in the context of chronic back pain, one known gastritis, one pneumonia, one uterine fibroma, and one possible viral rhinitis. The remaining 69 AEs, all considered possibly related to the intervention, were reported by 54 participants. Among these, 41 participants experienced a single AE, 11 experienced two AEs, and 2 participants reported three AEs.

The AEs tended to occur earlier in the LEV-3 and LEV-5 groups than in the PLA group (Table 2). Among the 54 participants who reported at least one AE, 18 were in the PLA group, 24 in the LEV-3 group, and 12 in the LEV-5 group. A significantly lower number of AEs was reported in the LEV-5 group ($P = 0.047$). No serious adverse events (SAEs) or grade 3 or 4 events were recorded, and there was no significant difference in AE severity across the three groups ($P = 0.999$).

The most frequently reported symptoms were asthenia, pruritus, and headaches (Supplementary Information. Table 1). Differences were observed between groups: gastrointestinal and skin disorders were more common in the PLA group. Furthermore, the LEV-3 group exhibited a higher frequency of general disorders (mainly isolated asthenia and flu-like syndromes), as well as nervous and musculoskeletal disorders, compared to the other groups. Lastly, eye disorders were more prevalent in the LEV-5 group.

A logistic regression model was applied to identify factors associated with the occurrence of an AE (Supplementary Information. Table 2). The Cramer V test yielded a low value (0.1618), allowing the inclusion of both *L. loa* MFD at D1 and the percentage reduction of MFD at D3 in the same model. Age, sex, and *L. loa* MFD categories at D1 were not significantly associated with AEs. The LEV-5 group had an almost fivefold lower risk of AEs (adjusted odds ratio [aOR] 0.21 [95% CI 0.06–0.85], $P = 0.029$), while no significant association was found for the LEV-3 group. Finally, individuals with intermediate and high MFD reduction rates (≥50%) had increased odds of experiencing an AE: aORs of 3.94 (95% CI 1.02–15.26, $P = 0.047$) and 3.19 (95% CI 0.78–13.19, $P = 0.108$), respectively.

### Efficacy assessment
Considering the mITT analysis, both the median MFDs and the median reduction in *L. loa* MFDs were significantly different among the three arms at all post-treatment time points (Table 3) except for the median MFD at D30 ($P = 0.073$). In the LEV-3 arm, the reduction in *L. loa* MFD, compared to the PLA arm, significantly persisted until D7, appeared to diminish by D15), and was no longer significant by D30. In contrast, in the LEV-5 arm, *L. loa* MFD exhibited a marked and sustained significant reduction up to D30 compared to the PLA arm. A comparison between the two LEV arms shows that the median reduction rates in *L. loa* MFD were higher in the LEV-5 group than in the LEV-3 group at both D5 and D7: 51.4% versus 29.3% at D5 ($P = 0.048$) and 34.4% versus 24.3% at D7 ($P = 0.035$), respectively. Fig. 2 illustrates the changes in median and geometric mean MFDs in the three arms.

Table 4 presents the proportion of individuals with more than 40% and more than 80% reduction in *L. loa* MFD, relative to pre-treatment values, for each of the 3 treatment arms at D3, D5, D7, D15,

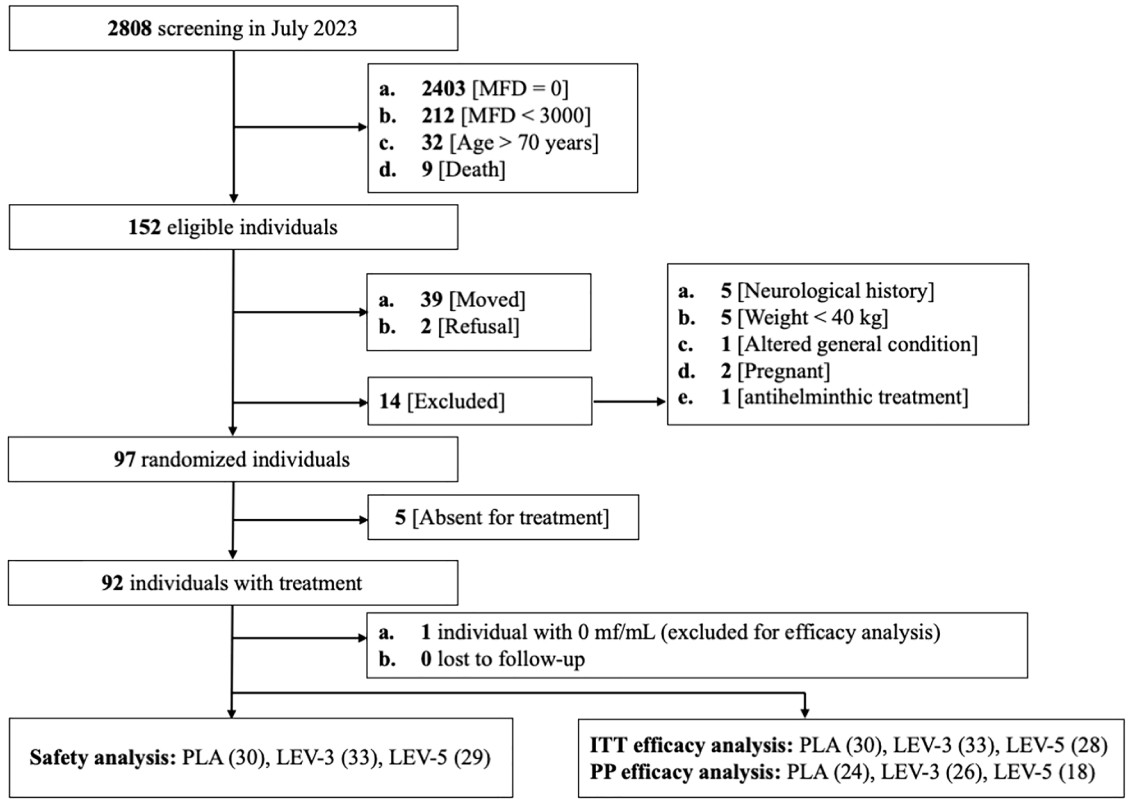

**Fig. 1 | Flowchart of screening and inclusion procedure.** MFD microfilarial densities, PLA placebo, LEV-3 3 days treatment with levamisole, LEV-5 5 days treatment with levamisole, ITT intention-to-treat, PP per protocol.

**Table 1 | Baseline (pre-treatment) characteristics of trial participants for all 92 participants**

|  | PLA (*N* = 30) | LEV-3 (*N* = 33) | LEV-5 (*N* = 29) |
|---|---|---|---|
| Sex |  |  |  |
| Female | 10 (33.3) | 10 (30.3) | 10 (34.5) |
| Male | 20 (66.7) | 23 (69.7) | 19 (65.5) |
| Age, mean ± SD | 50.1 ± 11.2 | 46.6 ± 13.1 | 47.0 ± 13.1 |
| MFD, mf/mL |  |  |  |
| Arithmetic mean ± SD | 24,796 ± 27,535 | 16,057 ± 13,168 | 20,594 ± 28,314 |
| Minimum; maximum | 4240; 113,500 | 15; 48,935 | 0/400ᵃ; 130,230 |
| Geometric mean (95% CI) | 16,268 (11,721–22,579) | 8883 (4974–15,863) | 11,607 (7311–18,426) |
| Median [IQR] | 13,475 [7780–23,805] | 12,775 [5695–24,675] | 10,850 [6335–21,690] |
| *Mansonella perstans* prevalence, N; % | 4; 13.1 | 1; 3.0 | 3; 10.3 |
| Heart rate, mean (bpm) ± SD | 75 ± 13 | 79 ± 18 | 77 ± 12 |
| Mean blood pressure, mean (mmHg) ± SD | 102 ± 15 | 102 ± 18 | 103 ± 16 |
| Systolic blood pressure, mean (mmHg) ± SD | 138 ± 22 | 137 ± 21 | 139 ± 22 |
| Diastolic blood pressure, mean (mmHg) ± SD | 84 ± 13 | 84 ± 17 | 85 ± 15 |
| Body temperature, mean (°C) ± SD | 36.1 ± 0.4 | 36.2 ± 0.4 | 36.2 ± 0.5 |

CI confidence interval, bpm beats per minute, IQR interquartile range, SD standard deviation, MFD Microfilarial density.
ᵃ 400 correspond to the minimum MFD after excluding the participant with 0 mf/mL.

and D30. The LEV-3 and LEV-5 arms demonstrated significant increases in these proportions compared to the placebo arm, with the LEV-5 arm showing more sustained effects. Additionally, a gradual effect of LEV-3 and LEV-5 was observed compared to the PLA group for the evaluation at D5 (PLA = 10.0, LEV-3 = 39.4, and LEV-5 = 67.9%, *P* between LEV-3 and LEV-5 = 0.024) and at D7 (PLA = 8.3, LEV-3 = 34.6, and LEV-5 = 44.4%, *P* = 0.545) for the 40% reduction threshold, as well as at D7 (PLA = 0.0, LEV-3 = 7.7, and LEV-5 = 16.7%, *P* = 0.772) for the 80% reduction threshold.

Our multivariable mixed model over time showed that the interaction between time and treatment groups was significant (*P* < 0.001), and model fit was substantially improved when including a random slope for time (Akaike and Bayesian information Criteria–AIC 4906 and BIC 5017) compared to models without a random slope for time (AIC 4955 and BIC 5057). Age and sex were not significantly associated with changes in *L. loa* MFD over time (Supplementary Information. Table 3). In contrast, higher baseline *L. loa* MFD was significantly associated with higher post-treatment values. Post hoc analyses

**Table 2 | Description of the adverse events (AEs) possibly related to treatment, and of the subjects having developed such AEs in the three treatment arms**

| Adverse events (AE) | PLA (N = 30) | LEV-3 (N = 33) | LEV-5 (N = 29) | P value |
|---|---|---|---|---|
| Total No. of AEs | 23 | 29 | 17 | |
| No. (%) of subjects with at least one AE | 18 (60.0) | 24 (72.7) | 12 (41.4) | 0.047[a] |
| Median onset, days [b] (range) | 1.5 (0–12) | 0 (0–3) | 1 (0–12) | 0.132[c] |
| Mean onset, days [b] (SD) | 2.2 (3.4) | 0.7 (1.1) | 1.9 (3.3) | |
| Median duration, days [b] (IQR) | 0.5 (0–5) | 0 (0–3) | 2 (0–10) | 0.182[c] |
| Severity of AEs | | | | |
| Grade 1 AE | 13 (72.2) | 17 (70.8) | 9 (75.0) | 0.999[a] |
| Grade 2 AE | 5 (27.8) | 7 (29.2) | 3 (25.0) | |
| Grade 3 & 4 AE | 0 | 0 | 0 | |

IQR interquartile range, SD standard deviation.
[a] Fisher's exact test calculated only on individuals who have reported an AE.
[b] Clinical AEs were reported up to 30 days.
[c] Kruskall-Wallis non-parametric test.

**Table 3 | Median microfilarial density (MFD), and median relative difference in MFD between DX (X = 3, 5, 7, 15, or 30) and D1 (pre-treatment) by arm (intention-to-treat analysis)**

| | Arms | | | P | | | |
|---|---|---|---|---|---|---|---|
| | **Median and Interquartile Ranges (IQR) of _L. loa_ MFD (mf/mL)** | | | | | | |
| Day | PLA (N = 30) | LEV-3 (N = 33) | LEV-5 (N = 28) | a | b | c | d |
| D1 | 13,475 (7780; 23,805) | 12,775 (5695; 24,675) | 11,817 (6882; 24,635) | 0.613 | 0.549 | 0.422 | 0.474 |
| D3 | 12,737 (8098; 35,570) | 7430 (3290; 12,990) | 6,942 (3982; 13,905) | 0.004 | 0.004 | 0.007 | 0.412 |
| D5 | 15,195 (7500; 24,725) | 9555 (3745; 13,670) | 5,227 (2527; 10,555) | 0.001 | 0.009 | <0.001 | 0.145 |
| D7 | 15,187 (9567; 30,832) | 9940 (3855; 12,645) | 5,902 (1985; 17,520) | 0.009 | 0.021 | 0.006 | 0.207 |
| D15 | 13,315 (8020; 32,215) | 10,515 (5315; 19,030) | 7,927 (5127; 16,935) | 0.050 | 0.072 | 0.029 | 0.265 |
| D30 | 13,802 (7065; 37,390) | 13,600 (6030; 20,555) | 7,287 (5272; 15,102) | 0.073 | 0.131 | 0.033 | 0.215 |
| | **Median and IQR of the reduction of _L. loa_ MFD compared to the pre-treatment result (%)** | | | | | | |
| Day | PLA (N = 30) | LEV-3 (N = 33) | LEV-5 (N = 28) | a | b | c | d |
| D3 | 0.4 (−9.4; 19.3) | 35.8 (19.1; 52.8) | 34.4 (23.3; 48.3) | <0.001 | <0.001 | <0.001 | 0.496 |
| D5 | −1.6 (−18.5; 16.7) | 29.3 (11.7; 58.1) | 51.4 (20.0; 63.5) | <0.001 | <0.001 | <0.001 | 0.048 |
| D7 | 8.2 (−13.2; 19.0) | 24.3 (12.1; 51.0) | 34.4 (25.5; 62.9) | <0.001 | 0.004 | <0.001 | 0.035 |
| D15 | −10.1 (−35.8; 14.9) | 16.0 (−26.2; 38.3) | 15.4 (5.4; 41.0) | 0.009 | 0.039 | 0.004 | 0.149 |
| D30 | −1.3 (−34.7; 20.9) | 2.0 (−20; 35.7) | 17.7 (0.5; 45.9) | 0.040 | 0.149 | 0.017 | 0.119 |

Kruskal-Wallis non-parametric test: (a) between the three arms; (b, c and d) _post hoc_ analyses with Dunn's pairwise test using a Holm correction for multiple tests: (b) between LEV-3 and PLA arms, (c) between LEV-5 and PLA arms, and (d) between LEV-5 and LEV-3 arms.

(Table 5) revealed that, beyond the significant effects of the LEV-3 and LEV-5 groups compared to PLA, the LEV-5 group was significantly more effective than the LEV-3 group. Specifically, as illustrated in Fig. 3, individuals in the LEV-5 group showed a 14.9% (P = 0.053) and 23.8% (P = 0.006) greater reduction in pre-treatment _L. loa_ MFD at D5 and D7, respectively, compared to those in the LEV-3 group.

Individual _L. loa_ MFD trajectories (Fig. 4) illustrate low inter-individual variability in the LEV-5 group, despite two individuals in the >20,000 mf/mL category who showed a rapid re-increase in MFD after Day 3–5. Overall, these individual trajectories were generally homogeneous, particularly during the first week in the intervention groups. In addition, we estimated an intraclass correlation coefficient (ICC) of 51.3% (95% CI 38.8–63.6) for the full model. However, when the model was restricted to the first week of follow-up, the intraclass correlation coefficient (ICC) dropped to 5.7% (95% CI 0.7–33.2). This finding suggests very low inter-individual variability in response to treatment during the first week, with greater variability appearing later.

Among participants with >30,000 mf/mL at D1, the proportions falling below 30,000 mf/mL by D5 were 28.6% (2 of 7) in PLA, 60.0% (3 of 5) in LEV-3, and 50.0% (2 of 4) in LEV-5 (Fisher's exact test for LEV-3 vs LEV-5: P = 0.643). By D7, these proportions were 33.3% (2 of 6) in

the PLA group, 33.3% (1 of 3) in the LEV-3 group, and 50.0% (2 of 4) in the LEV-5 group (Fisher's exact test for LEV-3 vs LEV-5: P = 0.629) (Fig. 5).

Finally, PP analysis (Supplementary Information. Tables 4 and 5) showed results consistent with those of the mITT analysis, although the significant difference for comparisons of reduction percentages at 80% were not confirmed.

## Discussion

This clinical trial corroborates the findings of our previous pilot study[13], demonstrating that a short course of LEV achieves a significant, albeit transient, reduction in _L. loa_ MFD without posing a risk of SAEs.

The LEV-3 and LEV-5 arms exhibited similar efficacy profiles up to D3, after which the LEV-5 arm demonstrated greater reductions in _L. loa_ MFDs compared to the LEV-3 arm. Interestingly, this difference diminished after D15, with both groups showing similar MFD reduction by D30. This pattern confirms the transient nature of levamisole's effect and supports the hypothesis that its pharmacodynamics is dose-dependent, with higher cumulative doses resulting in a more sustained and robust suppression of circulating mf. Overall, these observations strongly suggest an absence of a direct microfilaricidal effect of LEV, as

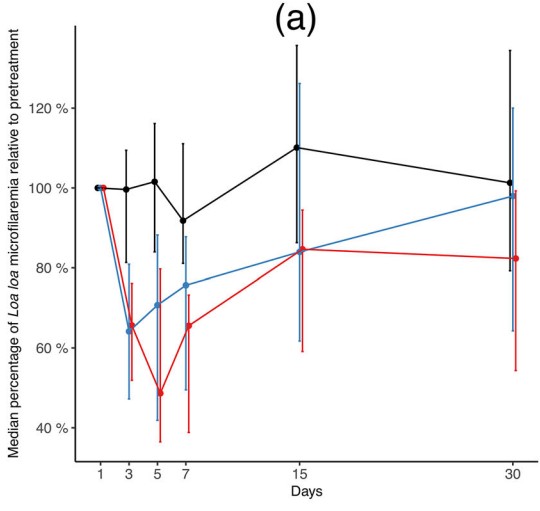

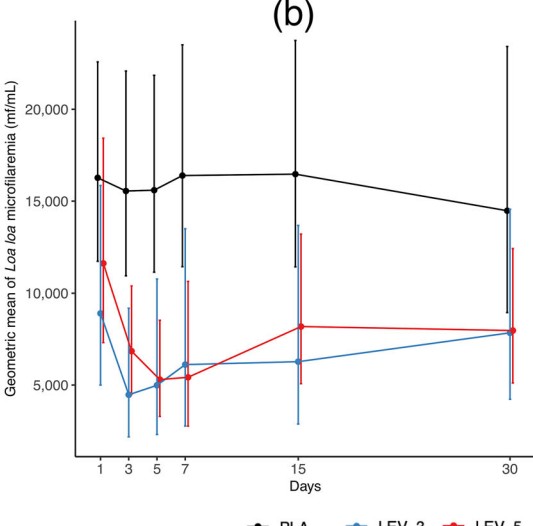

**Fig. 2 | Changes in *Loa loa* microfilarial densities in each treatment arm.** **a** Median and interquartile ranges. **b** Geometric means and their 95% confidence intervals. For (**a**) and (**b**), sample sizes used for each group were: PLA (*n* = 30), LEV-3 (*n* = 33), and LEV-5 (*n* = 28), except at Day 7, where the available sample sizes were PLA (*n* = 24), LEV-3 (*n* = 26), and LEV-5 (*n* = 18).

**Table 4 | Proportion of participants with a 40% and 80% reduction in their microfilarial density (MFD) per arm (intention-to-treat analysis)**

| | | Proportion of individuals with MFD reduction >40% | | | | | | |
|---|---|---|---|---|---|---|---|---|
| | | PLA | LEV-3 | LEV-5 | a | b | c | d |
| D3 | No | 28 (93.3) | 17 (51.5) | 16 (57.1) | | | | |
| | Yes | 2 (6.7) | 16 (48.5) | 12 (42.9) | <0.001 | <0.001 | 0.002 | 0.428 |
| D5 | No | 27 (90.0) | 20 (60.6) | 9 (32.1) | | | | |
| | Yes | 3 (10.0) | 13 (39.4) | 19 (67.9) | <0.001 | 0.014 | 0.004 | 0.024 |
| D7 | No | 22 (91.7) | 17 (65.4) | 10 (55.6) | | | | |
| | Yes | 2 (8.3) | 9 (34.6) | 8 (44.4) | 0.051 | 0.080 | 0.040 | 0.545 |
| D15 | No | 27 (90.0) | 25 (75.8) | 20 (71.4) | | | | |
| | Yes | 3 (10.0) | 8 (24.2) | 8 (28.6) | 0.372 | 0.372 | 0.280 | 0.462 |
| D30 | No | 28 (93.3) | 26 (78.8) | 17 (60.7) | | | | |
| | Yes | 2 (6.7) | 7 (21.2) | 11 (39.3) | 0.033 | 0.196 | 0.012 | 0.196 |
| | | Proportion of individuals with MFD reduction >80% | | | | | | |
| | | PLA | LEV-3 | LEV-5 | a | b | c | d |
| D3 | No | 30 (100.0) | 28 (84.8) | 27 (96.4) | | | | |
| | Yes | 0 | 5 (15.2) | 1 (3.6) | 0.136 | 0.136 | 0.483 | 0.280 |
| D5 | No | 30 (100.0) | 30 (90.9) | 26 (92.9) | | | | |
| | Yes | 0 | 3 (9.1) | 2 (7.1) | 0.687 | 0.548 | 0.687 | 0.687 |
| D7 | No | 24 (100.0) | 24 (92.3) | 15 (83.3) | | | | |
| | Yes | 0 (0.0) | 2 (7.7) | 3 (16.7) | 0.284 | 0.772 | 0.284 | 0.772 |
| D15 | No | 29 (96.7) | 30 (90.9) | 27 (96.4) | | | | |
| | Yes | 1 (3.3) | 3 (9.1) | 1 (3.6) | 0.988 | 0.878 | 0.999 | 0.678 |
| D30 | No | 29 (96.7) | 32 (97.0) | 26 (92.9) | | | | |
| | Yes | 1 (3.3) | 1 (3.0) | 2 (7.1) | 0.976 | 0.999 | 0.848 | 0.764 |

Two-sided Fisher's exact test with Holm correction for multiple test (adjusted-*P* values): (a) between the three arms, (b) between LEV-3 and PLA arms, (c) between LEV-5 and PLA arms, and (d) between LEV-5 and LEV-3 arms.

previously suggested by Mak et al.[24]. Instead, levamisole may induce paralysis of the mf, possibly through the inhibition of fumarate reductase activity[25] or the disruption of ion channels in their membranes[26]. This paralysis could impair their mobility, hindering their ability to maintain their presence in the bloodstream and promoting their sequestration, particularly in the lungs, where their ability to pass along the capillaries might be reduced. However, this sequestration does not affect all circulating mf, as evidenced by the significant number that remain detectable in the blood during treatment. Furthermore, the observation of mf with a more linear morphology under microscopic examination in the LEV-treated groups suggests a general effect of the drug on their structure and mobility. This is consistent with in vitro findings in *Brugia pahangi*, where levamisole-induced paralysis was associated with altered glucose

**Table 5 | Contrasts arm by arm for each time point of follow-up**

| Time | LEV-3 vs PLA | | LEV-5 vs PLA | | LEV-5 vs LEV-3 | |
| | Contrasts and 95% CI | P-values | Contrasts and 95% CI | P-values | Contrasts and 95% CI | P-values |
| --- | --- | --- | --- | --- | --- | --- |
| Day 1 | 0,50 [−13.6; 14.6] | 0.945 | −0.4 [14.3; −15.1] | 0.956 | −0.9 [13.4; −15.2] | 0.900 |
| Day 3 | −30.7 [−16.1; −45.4] | <0.001 | −31.9 [−16.7; −47.0] | <0.001 | −1.1 [13.6; −15.8] | 0.880 |
| Day 5 | −26.9 [−11.9; −41.9] | <0.001 | −41.8 [−26.2; −57.4] | <0.001 | −14.9 [0.2; −30.1] | 0.053 |
| Day 7 | −16.9 [−0.5; −33.2] | 0.043 | −40.7 [−23.2; −58.2] | <0.001 | −23.8 [−6.8; −40.9] | 0.006 |
| Day 15 | −12.9 [4.7; −30.5] | 0.152 | −23.3 [−4.9; −41.7] | 0.013 | −10.5 [7.3; −28.3] | 0.248 |
| Day 30 | −7.1 [15.4; −29.6] | 0.535 | −23.0 [0.4; −46.5] | 0.054 | −15.9 [6.9; −38.7] | 0.172 |

Pairwise comparisons using two-sided Wald tests. Exact p-values are reported. No correction for multiple testing was applied. 95% CI: 95% confidence interval.

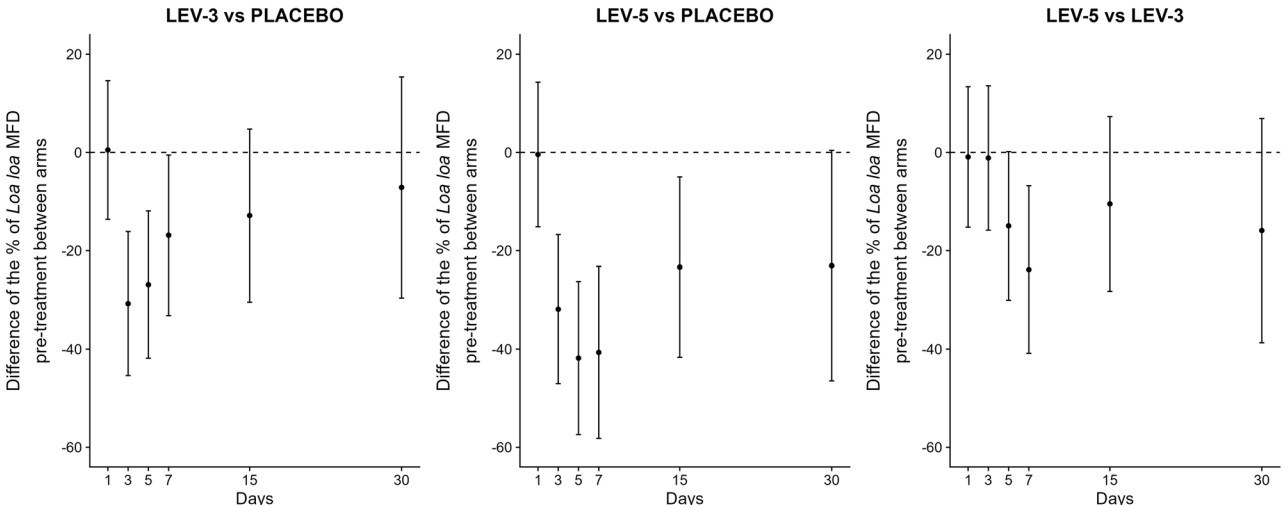

**Fig. 3 | Pairwise contrasts of pre-treatment percentage values between treatment arms (Arithmetic mean and their 95% confidence intervals).** Sample sizes used for each group were: PLA ($n = 30$), LEV-3 ($n = 33$), and LEV-5 ($n = 28$), except at Day 7, where the available sample sizes were PLA ($n = 24$), LEV-3 ($n = 26$), and LEV-5 ($n = 18$).

metabolism[27], and in *B. malayi*, where the drug caused rapid, reversible paralysis through nicotinic acetylcholine receptor activation[28].

Additionally, mf may be trapped in organs of the reticuloendothelial systems, such as the liver and the spleen, where their immobilization could enhance phagocytosis. Levamisole's immunomodulatory effect, known to potentiate macrophage activity, may play a complementary role in this process[24,29,30]. Nevertheless, the actual impact of this concurrent mechanism remains unclear, given the rapid onset of the treatment response and the short duration of LEV administration, which likely limits the extent of its immunomodulatory action.

A certain level of discrepancy exists between reductions in geometric means and percentage reductions, likely reflecting interindividual variability in treatment response. Although some deviations persist, examination of individual trajectories broadly indicates a high level of homogeneity. This variability appears to be even further reduced when examining only the first week, as suggested by our ICC estimates. For instance, one individual in the LEV-3 group experienced a marked increase in *L. loa* MFD, while two participants in the LEV-5 group—those with the highest initial MFDs—exhibited differing responses: one experienced a significant increase by D5 before completing the treatment course, and the other had a sharp rebound starting from D7. For the LEV-3 case, although a misallocation to the placebo group is theoretically possible, the observed increase in this LEV-3 group (Fig. 4 stratum <8000 mf/mL) markedly differs from the placebo profiles, making this hypothesis highly unlikely. Since LEV is known for its potential immunomodulatory effects[30], it could potentially interfere with the immunity established to regulate *L. loa* MFD, leading to therapeutic escape in some cases, as may be the situation

with our three participants. Further research would be needed to explore this hypothesis. However, as, on average, we found no significant association between baseline MFD and post-treatment MFD percentage reductions, it remains likely that if such an effect exists, it is merely anecdotal. Nonetheless, we regret not having been able to measure levamisole levels, which would have allowed for a better assessment of treatment response through pharmacokinetic analysis—an approach that might have revealed some inter-individual variability due to pharmacokinetic factors. Indeed, previous studies have reported significant inter-patient variability in levamisole pharmacokinetics[31,32], similar to what has been observed with other anthelmintics such as albendazole. Recognizing this variability is essential, as it may influence treatment efficacy and should be considered in future investigations.

In a systematic study of AEs associated with LEV anti-infective treatment, the most frequently reported effects were gastrointestinal, neurological, general, and dermatological[29]. Moreover, the AEs reported after LEV treatment in an anti-infective context were generally mild. This aligns perfectly with our findings. Although the absence of grade 3/4 events and SAEs is reassuring, the sample size was limited, reducing our ability to comprehensively assess safety. Differences in AEs were observed between treatment arms, but these varied across groups, complicating uniform interpretation. Notably, we confirmed that regardless of treatment arm or baseline MFD, individuals showing greater reductions in MFD tended to present more often with mild-to-moderate AEs.

Given its efficacy and safety, LEV could be incorporated into therapeutic recommendations, particularly for individualized patient care outside endemic regions and in specialized centers within endemic

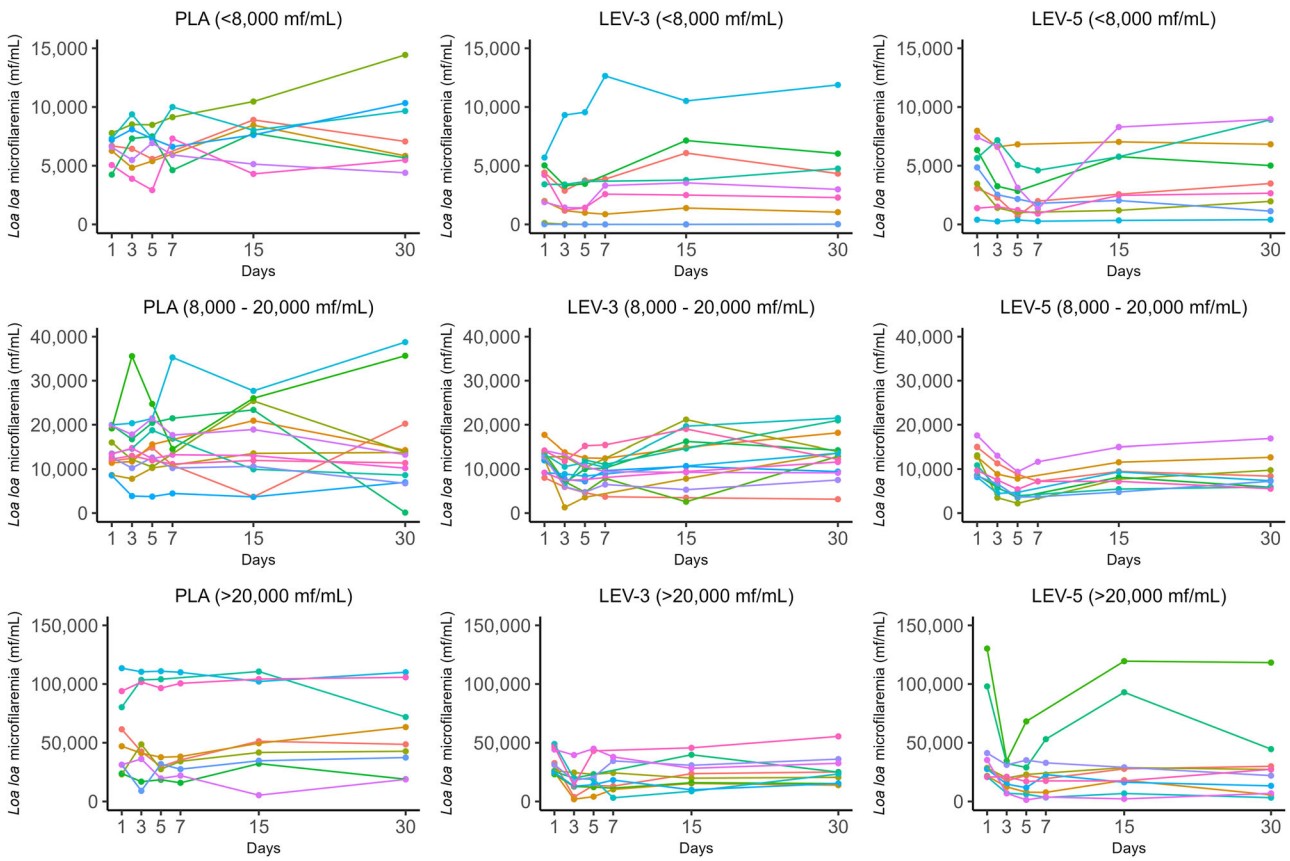

**Fig. 4 | Individual changes in *L. loa* microfilarial densities (MFD) by arm and by *L. loa* MFD category.** The three columns represent the three treatment arms of the study (from left to right: PLA, LEV-3, and LEV-5). The three rows represent microfilarial densities (from top to bottom: 8000, 8000–20,000, and 20,000 mf/mL).

areas. By D7, the proportions of participants achieving an MFD reduction exceeding 40% were 26.7%, 42.4%, and 64.3% in the PLA, LEV-3, and LEV-5 groups, respectively. For 80% reductions, these proportions were 20.0%, 27.3%, and 46.4%. Currently, the most commonly used protocol for patients with MFDs above 30,000 mf/mL, a key threshold for administering ivermectin in certain treatment protocols[33], is albendazole at 200 mg daily for three weeks. However, this regimen has no effect on *L. loa* MFD by Day 7[5]; and can lead to adverse hepatic effects, necessitating liver enzyme monitoring. Additionally, responses to albendazole are subject to high inter-individual variability, even by Day 30. Plasmapheresis is another option, but it requires hospitalization and incurs substantial costs, limiting its accessibility in many settings. In this trial, the proportion of participants transitioning from an MFD exceeding 30,000 mf/mL to a MFD below this threshold at D5 or D7 was similar in the LEV-3 and LEV-5 groups; although the small number of individuals with microfilarial densities above 30,000 mf/mL may account for the absence of statistical significance for this outcome. However, based on all of our results including the higher significant effect of LEV-5 vs LEV-3 at D5 and D7 in our mixed model, we recommend a 5-day course of LEV over a 3-day course. LEV offers a low-cost, without biology monitoring, short-course (5-day) alternative for individuals with high MFDs. In such cases, *L. loa* MFD should be reassessed at D7 before administering ivermectin. A second LEV course could be considered for non-responders.

While a 5-day LEV course appears most suitable for individual treatment scenarios, its potential application as a pretreatment strategy deserves further exploration. This is particularly relevant in areas where MDA for onchocerciasis is limited due to the risk of *Loa*-related post-IVM areas[2], where evaluating LEV-5 as a pretreatment option, may offer a new alternative therapeutic strategy. Ivermectin could then be safely administered one week after initiating LEV treatment. However, in this context, protocol adaptations (such as increasing the dosage, extending the duration, or combining with albendazole, for example) could be evaluated. Finally, in the context of a mass treatment program, systematic measurement of *L. loa* MFD between LEV and IVM administration would likely not be feasible. Regarding the potential risk of administering the two drugs 24–48 h apart, we believe this risk is negligible. Indeed, levamisole's plasma half-life is approximately 3–4 h, yielding essentially complete elimination within 24 h. If ivermectin is administered at least 24–48 h after the last levamisole dose, overlap of levamisole exposure is negligible, and CYP3A4 mediated metabolic interactions are unlikely. However, previous studies have reported clinically significant drug-drug interactions when levamisole and ivermectin are co-administered, including increased AUC and $C_{max}$ of ivermectin and reduced AUC and $C_{max}$ of albendazole sulphoxide[34]. Importantly, levamisole and ivermectin exert their effects on distinct parasite ion channels—nicotinic acetylcholine and glutamate-gated chloride channels, respectively—minimizing the likelihood of enhanced host toxicity. However, we therefore consider the risk of meaningful pharmacokinetic or pharmacodynamic interactions in our sequential dosing regimen to be minimal, provided a 24–48 h wash out is observed.

In conclusion, both 3-day and 5-day LEV regimens significantly but transiently reduce *L. loa* MFDs, with a favorable safety profile for individuals with high MFDs. A 5-day protocol may provide a practical solution for managing patients with MFDs exceeding 30,000 mf/mL before administering ivermectin. Further research should assess the role of LEV as a potential alternative strategy in onchocerciasis control programs in certain regions of Central Africa.

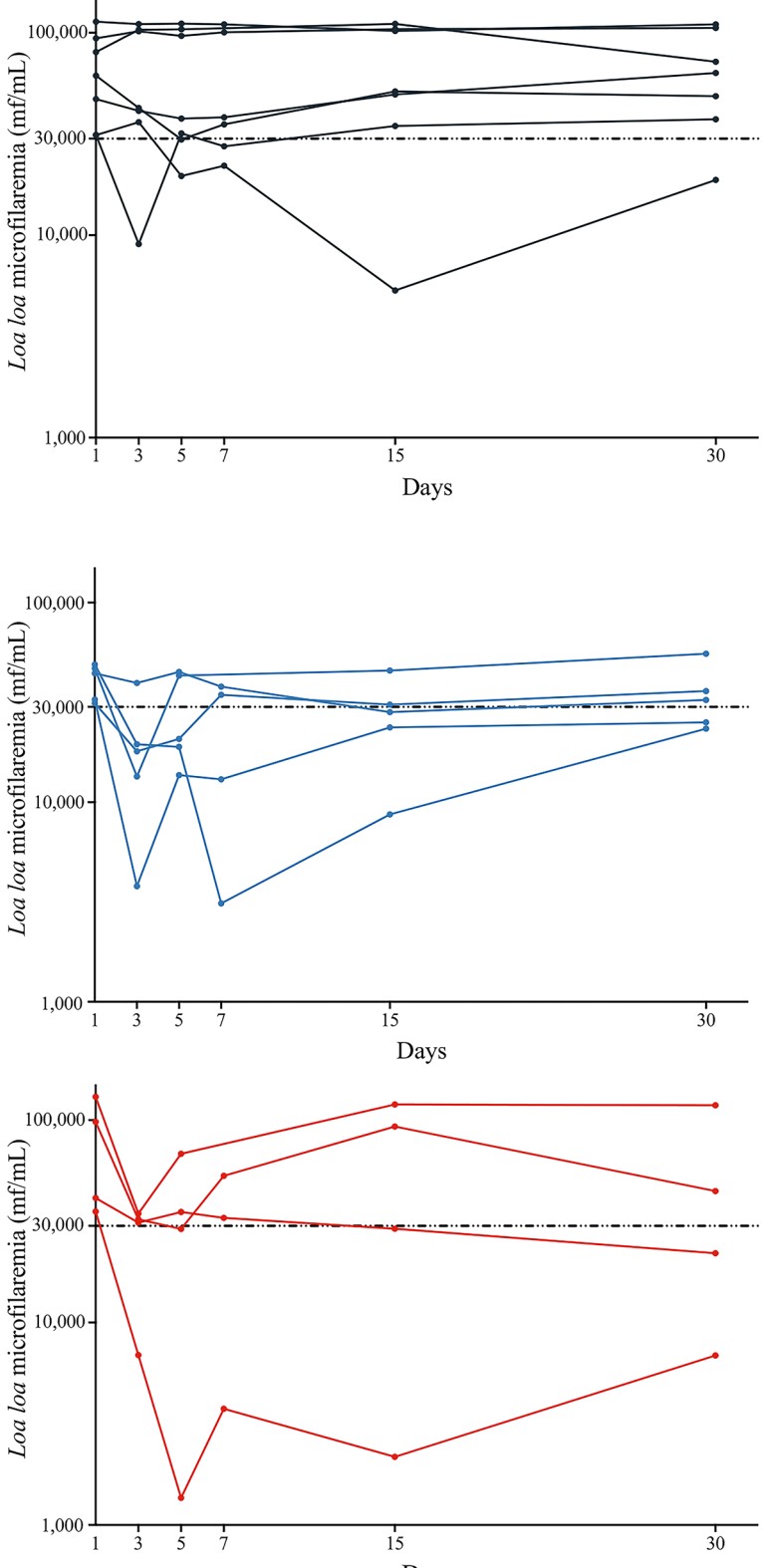

**Fig. 5 | Individual changes in microfilarial densities by arm in the participants with >30,000 mf/mL at baseline (D1).** Black: PLA group, Blue: LEV-3 group, and Red: LEV-5 group. Y-axis and X-axis: 10 logarithme scale.

## Methods

### Study area and selection of participants

Participants were recruited from 19 villages within 50 kilometers radius of Komono (3°16′39″S, 13°13′14″E), a small town located approximately 60 kilometers from Sibiti, the capital of the Lékoumou Division in the

Republic of Congo. This study site is located in a forested region where loiasis is endemic[35].

The recruitment process was carried out in two phases. Mid-2023, a population survey was conducted to screen for loiasis in the villages near Komono. In June 2024, individuals between 18 and 70 years old,

weighing between 40 and 90 kg and presenting with more than 3000 *L. loa* mf/mL during the first survey were invited for re-evaluation to determine their eligibility for the trial. This MFD threshold was arbitrarily set to ensure the individuals would still be microfilaremic at the start of the trial proper, given the relative stability of MFD over a one-year interval[36].

Volunteers underwent a thorough medical evaluation. Participants were excluded if they had taken part in any non-observational studies or received vaccinations within four weeks prior to the study. Those with acute infections requiring treatment within the past ten days or with a history of neurological or neuropsychiatric disorders, including epilepsy, were also excluded. Individuals using medications known to interact with LEV, such as clozapine or phenothiazines, within ten days before the study, or those with known immunosuppressive conditions or a history of agranulocytosis, were not eligible. Additionally, participants who used cocaine or other illicit drugs within 72 h before treatment, had known intolerance to LEV, or had donated more than 500 mL of blood in the past eight weeks were excluded. Any condition deemed to pose an undue risk by the investigator also led to exclusion. Additional exclusions included women pregnant for fewer than three months, detected systematically with urinary rapid diagnostic test, and those who had received an anthelminthic in the month preceding the trial. All microfilaremic individuals during the 2023 survey and fulfilling these criteria in 2024 were eligible for inclusion. The clinical trial was conducted from July to September 2024.

### Study design

For safety assessment, an independent Data Safety Monitoring Board reviewed the safety and efficacy results and could be consulted in the event of any unexpected clinical anomaly or SAE. Five days before treatment (D-5), all participants underwent a medical examination and completed a questionnaire to check for inclusion and exclusion criteria (see below). Once inclusion and non-inclusion criteria were met and the informed consent form was signed, participants were allocated to one of 3 arms: LEV 2.5 mg/kg during 3 days (LEV-3), LEV 2.5 mg/kg during 5 days (LEV-5), or placebo (PLA). Safety was checked until D30.

To assess efficacy, *Loa* MFD was measured just before the administration of the first tablet of treatment (D1), at day 3 (D3−48 h after the first dose), day 5 (D5−96 h after the first dose), at day 7 (D7), day 15 (D15), and at day 30 (D30) post-treatment. At D3, D5, and D7, each participant underwent a medical examination and screening for any adverse events (AEs). A medical team visited the villages of all participants every day from D2 to D7 to record and manage AEs. Conjunctivae were systematically examined, and if any abnormalities, febrile symptoms, or headaches were observed, a fundoscopic examination was conducted with a non-mydriatic retinal camera (Aurora, Optomed, DiTE sarl, Paris). All subjects received a participant card with emergency contact information.

### Randomization, blinding and drug preparation and administration

A 1:1:1 randomization of 3 arms (LEV-3, LEV-5, or PLA). with blocks size of 3 was performed. Randomization was carried out by an independent statistician, with stratification based on sex, age and *Loa* MFD. For the study, sealed containers were prepared, each containing the appropriate number of LEV tablets−based on the participant's weight−or matching placebo. Throughout the five-day treatment period, participants took their medication under the supervision of the trial's physicians. Individuals randomized to the LEV-5 arm received 5 days of LEV, those in the LEV-3 arm received LEV from D1 to D3 then placebo from D4 to D5, and those in the PLA arm received placebo for 5 days. All procedures were double-blinded. All tablets were sourced from ACE Pharmaceuticals BV in Zeewolde, The Netherlands.

### Laboratory procedures

*Loa* MFD were measured by examining two 50 µL calibrated blood smears (CBS1 and CBS2) on D1, D3, D5, D7, D15, and D30. To account for the diurnal fluctuation of *Loa* MFD[36], blood samples for CBS preparation were collected between 10:00 AM and 3:00 PM. For consistency, smears from each participant were prepared at the same hour for each time point (mean difference with D1 ± 2.8 min; with 90% of our samples having a difference <26 min). Blood was obtained by fingerprick and spread onto two labeled slides, which were then dried at room temperature, dehemoglobinized, and stained with Giemsa within four hours. Two experienced technicians examined independently each slide under 100 × magnification. If the MFD readings differed by more than 35% between the two microscopists, the slides were re-examined blindly. The arithmetic means of the MFDs from the four readings (CBS1 by both readers and CBS2 by both readers) were calculated and expressed in mf/mL for the analyses.

### Objectives and outcome measures

The primary objective of the trial was to evaluate the safety of multiple-dose LEV in individuals with *Loa* microfilaremia. The primary outcome measures were (i) the occurrence of an SAE and (ii) the frequency of AEs during the first month post-treatment.

The secondary objective was to assess the effect of LEV on *L. loa* MFD measured by: (i) the MFD reduction rates at D3, D5, D7, D15, and D30, (ii) the proportions of subjects with MFD reduction rates ≥40% and/or ≥80% at each time point post-D1, and (iii) the efficacy comparison between our arms. Reduction rates (%) were calculated as follows: ((MFD at D1)−(MFD at DX))/(MFD at D1) multiplied by 100 with X = 3, 5, 7, 15, or 30 days. Finally, we evaluated the proportion of individuals whose MFD fell below the thresholds of 30,000 mf/mL at D5 and D7.

### Sample size calculation

Although our primary objective was safety, the absence of any data on SAE risk from our previous clinical trial[13], and the fact that SAE risk in *L. loa*-microfilaremic individuals depends directly on the number of mf paralyzed and/or destroyed in the first 24−48 h post-treatment, led us to base our sample-size calculation on treatment efficacy. In that trial, 17.4% of participants who received a single 2.5 mg/kg dose of LEV showed a ≥40% reduction in their initial MFD two days after treatment, compared to 1.2% in the placebo group. Assuming an additive effect over a longer treatment period (with a 3-day treatment expected to be three times more effective than a single dose), we estimated that a 3-day treatment would reduce initial MFD by ≥40% in 52.2% of subjects, and a 5-day treatment would achieve this in 87.0% of subjects. Based on these estimates, we calculated that including 27 participants per treatment group would provide 80% power to detect these differences in MFD reduction. To account for a 20% loss to follow-up, we aimed to enroll a total of 99 subjects (33 per group).

### Statistical analysis

Safety was assessed using the proportions of participants with at least one AE. Proportions were tabulated by AE severity score (CTCAE grading scale version 5.0, see Supplementary Information. Text 1) and arms, and compared using a two-sided exact Fisher test. Then, we performed logistic regression adjusted on *L. loa* MFD at D1 (<20,000, 20,000−29,999, and >30,000 mf/mL−which are the same categories used in our efficacy analysis.), age, sex, treatment arm, and percentage of reduction of *L. loa* MFD at D3 (<25%, 25−49%, and ≥50%).

For the efficacy analyses, *L. loa* MFD as well as reduction rates of *L. loa* MFD were calculated and compared between arms at D3, D5, D7, D15, and D30, using a two-sided non-parametric Kruskal-Wallis test following by an adjusted *P* values using Dunn test applying a Holm correction for multiple tests. The proportions of participants with MFD reduction exceeding 40% and 80% were compared between arms with

two-sided Fisher's exact tests at D3, D5, D7, D15, and D30, using a Holm correction for multiple tests. We first performed a modified intention-to-treat (mITT) analysis—our primary analysis—and secondly, a per-protocol (PP) analysis. We used a mixed-effects linear model to assess the percentage of pre-treatment *L. loa* MFD retained after treatment. This dependent variable was calculated as: (MFD at DX/MFD at D1), multiplied by 100 with X = 3, 5, 7, 15 or 30 days. This formulation represents the proportion of the initial microfilarial load that remained at each follow-up time point. The model included treatment group, time point, and their interaction as fixed effects, with adjustments for age, sex, and baseline MFD category ( <20,000, 20,000–29,999, and >30,000 mf/mL). Random intercepts and slopes for time points were specified for each participant to account for within-subject correlations, using an unstructured covariance matrix. The interaction between time points and treatment group was assessed with a likelihood ratio test. Baseline *L. loa* MFD was categorized because the linearity test was not significant, and this categorization was validated against the continuous variable using AIC and BIC. After fitting the model, we estimated our ICC. Finally, post hoc contrasts were performed to compare treatment efficacy between pairs of groups at each time point, assessing the significance of these differences (applying a Wald's test). Finally, given the commonly used threshold of 30,000 mf/mL to determine when to initiate IVM treatment after an albendazole course, we reported the proportion of individuals with *L. loa* MFD >30,000 mf/mL before treatment who fell below this threshold at D5 and D7.

All statistical analyses were performed using Stata 18 (StatCorps LP, College Station, Texas, USA). The figures were created using R software (version 4.3.2).

## Trial registration and ethic statement

This study was approved by the Committee on Ethics of the Foundation for Medical Research in Congo (No. 51/CEI/FCRM/2024). An Administrative Authorization (No. 000056/MSP/DGSSSa/DPM-19) was released by the Ministry of Health and Population of the Republic of the Congo. The French National Commission on Informatics and Liberty (CNIL DR-2024-099) approved that the study protocol was ensuring compliance with data protection regulations. This study was conducted in accordance with the rules of Good Clinical Practices. All participants signed an informed consent form before initiation of any study-related procedure. This trial is registered as number NCT06252961 in https://clinicaltrials.gov/.

## Reporting summary

Further information on research design is available in the Nature Portfolio Reporting Summary linked to this article.

## Data availability

The data generated in this study have been deposited in the DataSuds database under CC-BY license (https://doi.org/10.23708/TTJXSF). The data are openly available, access can be obtained by submitting a request to DataSuds repository.

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

## Acknowledgements

We acknowledge the surveyed population for screening in 2023; and the participants to this clinical trial included in 2024. We also acknowledge all the local personnel in Sibiti: Stanislas Madzou (nurse); Dr. Gayila (physician); Pauline Kombi, Synthia Ntsoumou, Christiane Abana, Albertine Pika, and Odette Sandji (field technicians); Vincent Moussounda, and Anicet Madoulou (drivers). We would also like to thank the late Mr. Pandzou, a field health community agent who died in 2024. Last, we are very grateful for the help of the personnel of the 'Secteur Opérationnel' of the Lekoumou Department for their support during this study as well as the local authorities. We would like to thank Dr. Thibaut for proofreading the English of the manuscript and Mr. Delmas for carrying out the randomization. The sponsor had no role in the study design, data collection and analysis of the present study. This work was supported by the French National Research Agency (ANR) (grant number 18-CE17-0008), granted by J.T.C.

## Author contributions

C.B.C., S.D.S.P., M.B., and J.T.C. participated in the conception and design of the study; M.C.H., M.A.N.N., M.P., and F.M. conducted the screening survey in 2023; C.B.C., M.C.H., B.A.S., F.T., E.D., N.N.E., M.A.N.N., M.P., F.M., J.T.C. carried out the clinical trial in 2024; C.B.C., B.A.S., N.N.E., and J.T.C. participated in the acquisition of data; C.B.C. performed the statistical analyses and wrote the first version of the manuscript; B.C. helped to statistical analyses and performed figures; E.D. supervised the pharmacy and randomization procedure for tablets. All authors approved the final version for publication.

## Competing interests

The authors no conflicts of interest. All authors have submitted the ICMJE Form for Disclosure of Potential Conflicts of Interest.
