## [Peer Review file · Nature Communications]

Safety and efficacy of 3- and 5-day regimens of levamisole in loiasis: a randomized, placebo-controlled, double-blind clinical trial

Corresponding Author: Dr Jérémy Campillo

Version 0:

Reviewer comments:

Reviewer #1

(Remarks to the Author)

This study highlights the potential of levamisole as a safer alternative for managing high Loa loa microfilarial densities, especially in regions where ivermectin poses a risk of serious adverse events (SAEs). The randomized, placebo-controlled trial conducted in the Republic of the Congo showed that both 3-day and 5-day courses of LEV significantly reduced MFDs without triggering SAEs or worsening adverse event profiles compared to placebo. Notably, the 5-day LEV regimen demonstrated the most substantial reduction, with over 64% of participants achieving at least a 40% drop in MFD by Day 7. The authors suggest that extended LEV dosing may offer a safer, effective pretreatment strategy for individuals at high risk of ivermectin-induced complications, and could support future individualized treatment approaches in coendemic areas where both loiasis and onchocerciasis overlap. Such studies are difficult to conduct and any information on treatment options for loiasis are highly relevant.

Comments:

Discussion:

- The discrepancy between the geometric mean reduction and the percentage of participants achieving specific microfilarial density reductions may reflect inter-individual variability in treatment response, but could also be potentially influenced by differences in pharmacokinetics of levamisole or natural fluctuations in parasite levels. This should be considered.
- While PK was not assessed in this study—an understandable limitation given the associated costs—a brief discussion of known inter-patient variability in levamisole exposure could enhance the reader's understanding. For instance, other anthelmintics like albendazole show significant PK variability, which may influence efficacy.
- A summary of preclinical (in vitro) evidence regarding levamisole's effects on Loa loa (or other) microfilariae morphology and mobility would provide helpful mechanistic insight into its mode of action.
- If levamisole is being considered as a pre-treatment strategy before ivermectin administration, it would be beneficial to discuss its half-life, potential drug-drug interactions (DDIs), and how these may affect its suitability in such a individual treatment application.
- Including a "limitations of the study" paragraph would strengthen the discussion. Key points could include the absence of levamisole PK data, the follow-up period (which limits conclusions about the sustainability of MFD reduction).

Methods:

- Please clarify why the smear evaluation threshold was set at 35% and explain how this cutoff could influence the interpretation of efficacy outcomes.
- While the rationale for basing the sample size on efficacy data is clear, why the safety was the primary objective but not the efficacy. If the focus was safety, a justification for not incorporating drug-induced (rather than efficacy-induced) adverse events (as per levamisole regulatory studies) into the calculation would improve clarity and alignment with the study's objectives.

Reviewer #2

(Remarks to the Author)

This study was designed to assess 3 and 5 days of levamisole for the treatment of loiasis in Republic of Congo. This study is a follow-up study from a previous trial evaluating single dose levamisole treatment. Only few clinical trials have been conducted in the past decades for loiasis leading to only limited progress in the development of new drugs and drug regimens. Therefore this study is highly welcome.

The study was performed diligently and results are reported in a balanced way. The primary outcome was safety and tolerability. Given the small sample size, these data are of importance but likely not conclusive on the potential for rare, serious adverse drug reactions including encephalitis. The pharmacodynamic characterization is of interest and indicates a modest and only transient effect. The follow-up period was short with 30 days but data indicate that the reduction of microfilaremia is short-lived and longer follow-up would therefore be of limited interest.

The authors may be asked whether the conclusion of levamisole being a promising drug for individual patient management is justified given the only modest reduction of microfilarial load, the only short-lived effect on peripheral microfilaraemia and the unknown effect on the safety of subsequent treatment with a rapidly acting antifilarial drug.

Reviewer #3

(Remarks to the Author)

This is a randomized controlled trial evaluating two new regimens of levamisole in comparison to a placebo control for the treatment of individuals with *L. loa* MFD. This is an important trial that is constructed appropriately and the paper is generally well-written. The findings do support the conclusion that both 3 and 5-day regimens of LEV are safe and effective. However, I have many concerns related to the presentation of the results, some of the conclusions drawn, and the methods. Specific comments are noted below, but in particular I do not believe the current presentation of results support the claim that the 5-day LEV protocol is best particularly for individuals with $>30,000$ mf/mL and I do not understand why the authors did not use repeated measures models to evaluate changes in MFD over time.

Abstract

- Several key pieces of information are not included in the abstract, namely a brief mention of statistical methods used and the study's sample size
- The following conclusion stated in the abstract does not appear to be supported by the results presented: "A 5-day course of LEV appears to be an option for the individualised management of patients with *L. loa* MFD exceeding 30,000 microfilariae per mL of blood."

Results

- It seems that the 1 individual with 0mf/mL *L. loa* MFD should probably be excluded from all analyses (safety and efficacy)?
- Please also include %s for Sex in Table 1
- SDs should always be reported alongside means (line 113) and IQRs should always be reported alongside medians (lines 119-124)
- Depending on which is considered the primary analysis of interest, I would recommend including information on either the mITT or PP analyses in the results section, as opposed to both. Whichever is not included in the main body of the text can be moved to supplemental material.
- As the methods section states that safety is the primary outcome of interest for this paper, the safety results should be reported first, followed by the efficacy results
- You may consider restructuring tables 2 and 3 to only include the p-value for the overall three-group comparison and then include superscripts to denote the significant pairwise comparisons.
- No need to report p-values for non-significant variables (lines 147-150), particularly given that effect estimates are also not reported.
- The purpose of figure 3 and why the information shown is critical to the current study is not clear to me
- Why did you examine participants with *L. loa* MFD $> 30,000$? No justification is provided and this subgroup analysis is not described in the methods section
- The sentence in lines 186-188 needs to be revised for clarity.
- Does Table 5 show the number of unique AEs of each type or the number of unique subjects reporting each AE type?
- Full results from the logistic regression model predicting AE occurrence (lines 208-216) should be reported in a table

Discussion

- The description of results from Figure 3 (lines 252-260) should be presented first in the Results section and then commented on in the Discussion section
- I do not see where the following statement on lines 287-289 is substantiated in the Results: "In this trial, the proportion of participants 288 transitioning from an MFD exceeding 30,000 mf/mL to an MFD below this threshold at D5 or 289 D7 was similar in the LEV-3 and LEV-5 groups"
- Similarly, and in line with my comment on the abstract, I do not see how the results support the authors' conclusion that the 5-day protocol is best for individuals with $> 30,000$ mf/mL MFD.

Methods

- I assume the formula for calculating reduction rates should include multiplying the proportion by 100?
- The purpose of the logistic regression as described in lines 407-410 is unclear. Also, why was *L. loa* MFD categorized in this way instead of using the continuous value or the same categorizations for the efficacy analysis? It is also unclear

- whether controlling for percentage reduction of MFD at D3 would be appropriate if AEs could have occurred prior to D3?
- Lines 416-417 indicate that both mITT and PP analyses were conducted. Which can be considered the primary analysis?
 - Were additional individual characteristics (age, sex, treatment) controlled for in the linear regression described in lines 417-420? Based on the results it seems that they were, but this needs to be described in the methods as well.
 - Why wasn't a repeated measures model used to examine changes in MFD over time?

Version 1:

Reviewer comments:

Reviewer #1

(Remarks to the Author)

Reviewer #3

(Remarks to the Author)

I commend the authors for their careful revision of this manuscript. The majority of my prior concerns have been adequately addressed. I just have a few minor comments remaining.

- The revised sentence on lines 131-133 is still odd. Inherently by nature of the fact that 54 participants reported 69 AEs, there must have been multiple AEs reported by some individuals, so the phrase "with some participants experiencing more than one AE" is extraneous. It might instead be useful to know the variability in how many AEs were reported total.
- In Table 3, reporting the frequency of AEs but the percentage of individuals is very confusing. I would stick with one or the other, either the number of unique individuals reporting that AE category and associated % of individuals or number of unique AEs and % out of total AEs.
- Please explicitly state the how the dependent variable for the linear mixed effects model ("percentage of pre treatment L. loa MFD retained after treatment") was calculated, as this appears to be different from the reduction rate specified in the "Objectives and outcome measures" section.
- The acronyms AIC, BIC, and ICC should be spelled out prior to first use.

REVIEWER COMMENTS

Reviewer #1 (Remarks to the Author):

This study highlights the potential of levamisole as a safer alternative for managing high *Loa loa* microfilarial densities, especially in regions where ivermectin poses a risk of serious adverse events (SAEs). The randomized, placebo-controlled trial conducted in the Republic of the Congo showed that both 3-day and 5-day courses of LEV significantly reduced MFDs without triggering SAEs or worsening adverse event profiles compared to placebo. Notably, the 5-day LEV regimen demonstrated the most substantial reduction, with over 64% of participants achieving at least a 40% drop in MFD by Day 7. The authors suggest that extended LEV dosing may offer a safer, effective pretreatment strategy for individuals at high risk of ivermectin-induced complications, and could support future individualized treatment approaches in coendemic areas where both loiasis and onchocerciasis overlap. Such studies are difficult to conduct and any information on treatment options for loiasis are highly relevant.

Response: We thank the reviewer for this comment.

Comments:

Discussion:

- 1) The discrepancy between the geometric mean reduction and the percentage of participants achieving specific microfilarial density reductions may reflect inter-individual variability in treatment response, but could also be potentially influenced by differences in pharmacokinetics of levamisole or natural fluctuations in parasite levels. This should be considered.

Response: We thank the reviewer for raising the issue of possible inter-individual variability in treatment response. As noted below (see point 2), we were unable to perform levamisole pharmacokinetic assays due to budgetary constraints, and so could not directly account for pharmacokinetic sources of response variability. In response to Reviewer 3, we have now fitted a mixed-effects model to our trial data, which allows us both to estimate the intraclass correlation coefficient (ICC) and to quantify inter-individual variability in treatment response. The estimated ICC was very low (approximately 5%) during the first week of follow-up, indicating that inter-subject variability in response is minimal during this period. Any observed variability in treatment effect appears to arise largely from within-subject factors—primarily whether the participant received active drug versus placebo, the post-treatment time point, and potentially other unmeasured covariates—rather than from stable differences between individuals. In other words, the levamisole effect is relatively homogeneous across patients for one week. We have added these results and interpretation to the Discussion, including: (i) a statement of the estimated ICC and its interpretation regarding the low inter-individual variability; and (ii) a commentary on how baseline microfilarial density influenced the magnitude of reduction over time, which may explain the apparent discrepancy between geometric-mean reductions and the proportion of individuals achieving $\geq 40\%$ or $\geq 80\%$ reduction.

- 2) While PK was not assessed in this study—an understandable limitation given the associated costs—a brief discussion of known inter-patient variability in levamisole exposure could enhance the reader's understanding. For instance, other anthelmintics like albendazole show significant PK variability, which may influence efficacy.

Response: We thank the reviewer for raising this important point. Indeed, we had originally planned to perform levamisole assays in this clinical trial in order to carry out a pharmacokinetic evaluation. As noted, however, budgetary constraints ultimately prevented us from doing so. This limitation is particularly relevant

given the known inter-individual variability in levamisole pharmacokinetic,^{1,2} similar to what has been observed with other anthelmintics like albendazole. While our study represents the first evaluation of levamisole's effect on *Loa loa*, acknowledging this variability is important, as it may influence treatment efficacy. We have now added this consideration to the Discussion.

“Nonetheless, we regret not having been able to measure levamisole levels, which would have allowed for a better assessment of treatment response through pharmacokinetic analysis—an approach that might have revealed some inter-individual variability due to pharmacokinetic factors. Indeed, previous studies have reported significant inter-patient variability in levamisole pharmacokinetics [31,32], similar to what has been observed with other anthelmintics such as albendazole. Recognizing this variability is essential, as it may influence treatment efficacy and should be considered in future investigations.”

3) A summary of preclinical (in vitro) evidence regarding levamisole's effects on *Loa loa* (or other) microfilariae morphology and mobility would provide helpful mechanistic insight into its mode of action.

Response: No *in vitro* study has yet been published on levamisole's effects on the morphology or motility of *L. loa* microfilariae. Early work in *Onchocerca volvulus* demonstrated only a moderate impact of levamisole on embryogenesis in excised nodules and on *in vitro* development of that species' microfilariae.³ We therefore agree with the reviewer that dedicated *in vitro* investigations would be highly valuable to highlight levamisole's mechanism of action. Such work might adapt recently developed automated phenotype-tracking platforms (e.g. for *Brugia malayi*)⁴ to quantify, in a high-throughput way, changes in microfilarial motility and morphology under levamisole exposure. In support of this approach, studies in *Brugia pahangi* have shown that levamisole-induced paralysis is associated with altered aerobic glucose metabolism, including reduced glucose utilization and a shift toward homolactate fermentation.⁵ Similarly, in *B. malayi*, levamisole caused rapid, reversible paralysis of microfilariae *in vitro*, with an IC₅₀ of 2.68 mM after 2 hours, consistent with its action as a nicotinic acetylcholine receptor agonist.⁶ By contrast, existing clinical trials in other filariases have rarely reported detailed morphological endpoints. In *O. volvulus*, levamisole treatment produced reductions in microfilarial density of 30–44%,⁷ while in *Wuchereria bancrofti* (whose microfilariae also circulate in blood, albeit with nocturnal periodicity), reduction rates ranged from 78.8% to 98.5%.⁸ Importantly, in the present trial we observed—for the first

¹ Gwilt P, Tempero M, Kremer A, Connolly M, Ding C (2000). Pharmacokinetics of levamisole in cancer patients treated with 5-fluorouracil. *Cancer Chemother Pharmacol*, **45**(3):247-51. doi: 10.1007/s002800050036. PMID: 10663643.

² Luyckx, M., Rousseau, F., Cazin, M., Brunet, C., Cazin, J. C., Haguenoer, J. M., Devulder, B., Lesieur, I., Lesieur, D., Gosselin, P., Adenis L, Cappelaere P, Demaille A (1982). Pharmacokinetics of levamisole in healthy subjects and cancer patients. *European Journal of Drug Metabolism and Pharmacokinetics*, **7**(4), 247–254

³ Rivas-Alcala R, Mackenzie CD, Gomez-Rojo E, Greene BM, Taylor HR (1984). The effects of diethylcarbamazine, mebendazole and levamisole on *Onchocerca volvulus* in vivo and in vitro. *Tropenmed Parasitol*, **35**(2):71-7. PMID: 6464189.

⁴ Kalwa, U., Park, Y., Kimber, M.J. *et al* (2024). An automated, high-resolution phenotypic assay for adult *Brugia malayi* and microfilaria. *Sci Rep*, **14**, 13176.

⁵ Rew RS, Saz HJ (1977). The carbohydrate metabolism of *Brugia pahangi* microfilariae. *J Parasitol*, **63**(1):123-9. PMID: 191584.

⁶ Mostafa E, Storey B, Farghaly AM, Afify HA, Taha AA, Wolstenholme AJ (2015). Transient effects of levamisole on *Brugia malayi* microfilariae. *Invert Neurosci*, **15**(3):5. doi: 10.1007/s10158-015-0181-0.

⁷ Awadzi K, Schulz-Key H, Howells RE, Haddock DR, Gilles HM (1982). The chemotherapy of onchocerciasis VIII Levamisole and its combination with the benzimidazoles. *Ann Trop Med Parasitol*, **76**:459–73.; Rivas-Alcalá AR, Greene BM, Taylor HR, Domínguez-Vázquez A, Ruvalcaba-Macías AM, Lugo-Pfeiffer C, Mackenzie CD, Beltrán F (1981). Chemotherapy of onchocerciasis: a controlled comparison of mebendazole, levamisole, and diethylcarbamazine. *Lancet*, **318**: 485–90.

⁸ Merlin M, Carme B, Kaeuffer H, Laigret J. Activity of levamisole (Solaskil) in lymphatic filariasis caused by *Wuchereria bancrofti* (variety *pacifica*) (1976). *Bull Soc Pathol Exot Filiales*, 69:257–65. McMahon JE (1979). Preliminary screening of antifilarial activity of levamisole and amodiaquine on *Wuchereria bancrofti*. *Ann Trop Med Parasitol*, **73**:465–72. Narasimham MV, Roychowdhury SP, Das M, Rao CK (1978). Levamisole and mebendazole in the treatment of bancroftian infection. *Southeast Asian J Trop Med Public Health*, **9**:571–5. O'Holohan DR, Zaman V (1974). Treatment of *Brugia malayi* infection with levamisole. *J Trop Med Hyg*, **77**:113–5. Zaman V, Lal M (1973). Letter: Treatment of *Wuchereria bancrofti* with levamisole. *Trans R Soc Trop Med Hyg*, **67**: 610.

time— that microfilariae still present in the blood of levamisole-treated patients seemed to be more stretched than those found in the placebo group under microscopic examination, consistent with a direct structural and motility-impairing effect of the drug. We have added these points to the Discussion.

“This is consistent with *in vitro* findings in *Brugia pahangi*, where levamisole-induced paralysis was associated with altered glucose metabolism [27], and in *B. malayi*, where the drug caused rapid, reversible paralysis through nicotinic acetylcholine receptor activation [28].”

- 4) If levamisole is being considered as a pre-treatment strategy before ivermectin administration, it would be beneficial to discuss its half-life, potential drug-drug interactions (DDIs), and how these may affect its suitability in such an individual treatment application.

Response: We agree with the reviewer; this was also requested by reviewer 2. The following sentences were added in the revised manuscript: “Indeed, levamisole’s plasma half-life is approximately 3–4 hours, yielding essentially complete elimination within 24 hours. If ivermectin is administered at least 24–48 hours after the last levamisole dose, overlap of levamisole exposure is negligible, and CYP3A4 mediated metabolic interactions are unlikely. However, previous studies have reported clinically significant drug-drug interactions when levamisole and ivermectin are co-administered, including increased AUC and C_{max} of ivermectin and reduced AUC and C_{max} of albendazole sulphoxide [34].⁹ Importantly, levamisole and ivermectin exert their effects on distinct parasite ion channels—nicotinic acetylcholine and glutamate-gated chloride channels, respectively—minimizing the likelihood of enhanced host toxicity. However, we therefore consider the risk of meaningful pharmacokinetic or pharmacodynamic interactions in our sequential dosing regimen to be minimal, provided a 24–48 hours wash out is observed.”

- 5) Including a “limitations of the study” paragraph would strengthen the discussion. Key points could include the absence of levamisole PK data, the follow-up period (which limits conclusions about the sustainability of MFD reduction).

Response: We fully agree with the reviewer and the following paragraphs have been added:

“Nonetheless, we regret not having been able to measure levamisole levels, which would have allowed for a better assessment of treatment response through pharmacokinetic analysis—an approach that might have revealed some inter-individual variability due to pharmacokinetic factors.”

“Finally, in the context of a mass treatment program, systematic measurement of *L. loa* MFD between LEV and IVM administration would likely not be feasible. Regarding the potential risk of administering the two drugs 24–48 hours apart, we believe this risk is negligible. Indeed, levamisole’s plasma half-life is approximately 3–4 hours, yielding essentially complete elimination within 24 hours. If ivermectin is administered at least 24–48 hours after the last levamisole dose, overlap of levamisole exposure is negligible, and CYP3A4 mediated metabolic interactions are unlikely. However, previous studies have reported clinically significant drug-drug interactions when levamisole and ivermectin are co-administered, including increased AUC and C_{max} of ivermectin and reduced AUC and C_{max} of albendazole sulphoxide [34]. Importantly, levamisole and ivermectin exert their effects on distinct parasite ion channels—nicotinic acetylcholine and glutamate-gated chloride channels, respectively—minimizing the likelihood of enhanced host toxicity. However, we therefore consider the risk of meaningful pharmacokinetic or pharmacodynamic interactions in our sequential dosing regimen to be minimal, provided a 24–48 hours wash out is observed.”

⁹ Awadzi K, Edwards G, Opoku NO, Ardrey AE, Favager S, Addy ET, Attah SK, Yamuah LK, Quartey BT (2004). The safety, tolerability and pharmacokinetics of levamisole alone, levamisole plus ivermectin, and levamisole plus albendazole, and their efficacy against *Onchocerca volvulus*. *Ann Trop Med Parasitol*, **98**(6):595-614. doi: 10.1179/000349804225021370.

Methods:

- 6) Please clarify why the smear evaluation threshold was set at 35% and explain how this cutoff could influence the interpretation of efficacy outcomes.

Response: The smear evaluation threshold was set at 35% based on previous work we conducted, as there is no universally accepted repeatability coefficient recognized by accreditation instances for loiasis mfs count. For instance, the COFRAC document SH GTA 04 provides an accreditation example for microscopic malaria diagnosis, where a repeatability coefficient of 25.7% is considered acceptable. Regarding the coefficient of variation for readings, the average values recorded were 0.52%, 0.44%, 1.19%, 0.8%, 0.1%, and 2.42% for pre-treatment, Day 3, Day 5, Day 7, Day 15, and Day 30, respectively. These low variations confirm that the threshold selection does not influence the interpretation of efficacy results in any meaningful way.

- 7) While the rationale for basing the sample size on efficacy data is clear, why the safety was the primary objective but not the efficacy. If the focus was safety, a justification for not incorporating drug-induced (rather than efficacy-induced) adverse events (as per levamisole regulatory studies) into the calculation would improve clarity and alignment with the study's objectives.

Response: We thank the reviewer for this important point. Following our previous clinical trial,¹⁰ the excellent safety profile—and in particular the complete absence of any grade 3 or higher adverse events—meant that we had no reliable estimate of drug-induced adverse event rates on which to base a formal safety-focused sample-size calculation. We could instead have used a generic binomial approach—for example, calculating the number of participants needed to have an 80 % chance of observing at least one SAE event occurring at a given low incidence. However, we decided to link the risk of SAE to the kinetics of *L. loa* microfilarial clearance after treatment. Indeed, the risk of SAE in *L. loa* microfilaremic individuals is proportional to the pre-treatment number of microfilariae per mL of blood.¹¹ Therefore, we judged that basing our sample-size calculation on the observed treatment-response kinetics (efficacy) from our prior trial would meaningfully predict the risk of SAE—for example, an average 80 % reduction in MFD by day 3 after LEV treatment implies a correspondingly low risk of post-IVM SAE. For this reason, although our primary objective was safety, we calculated our required sample size using the efficacy parameters derived from our earlier study. We have now clarified this in the Methods section: “Although our primary objective was safety, the absence of any data on SAE risk from our previous clinical trial [13], and the fact that SAE risk in *L. loa*-microfilaremic individuals depends directly on the number of mf paralyzed and/or destroyed in the first 24–48 hours post-treatment, led us to base our sample-size calculation on treatment efficacy.”

Reviewer #2 (Remarks to the Author):

This study was designed to assess 3 and 5 days of levamisole for the treatment of loiasis in Republic of Congo. This study is a follow-up study from a previous trial evaluating single dose levamisole treatment. Only few clinical trials have been conducted in the past decades for loiasis leading to only limited progress in the development of new drugs and drug regimens. Therefore, this study is highly welcome.

The study was performed diligently and results are reported in a balanced way. The primary outcome was safety and tolerability. Given the small sample size, these data are of importance but likely not conclusive on the potential for rare, serious adverse drug reactions including encephalitis. The pharmacodynamic

¹⁰ Campillo, J. T., Bikita, P., Hemilembolo, M. C., Louya, F., Missamou, F., Pion, S. D. S., Boussinesq, M., & Chesnais, C. B. (2022). Safety and efficacy of levamisole in loiasis: a randomized, placebo-controlled, double-blind clinical trial. *Clinical Infectious Diseases*, 75(1), 19–27

¹¹ Chesnais, C. B., Pion, S. D. S., Boullé, C., Gardon, J., Kamgno, J., & Boussinesq, M. (2020). Individual risk of post-ivermectin serious adverse events in subjects infected with *Loa loa*. *EclinicalMedicine*, 28, 100582.

characterization is of interest and indicates a modest and only transient effect. The follow-up period was short with 30 days but data indicate that the reduction of microfilaraemia is short-lived and longer follow-up would therefore be of limited interest.

The authors may be asked whether the conclusion of levamisole being a promising drug for individual patient management is justified given the only modest reduction of microfilarial load, the only short-lived effect on peripheral microfilaraemia and the unknown effect on the safety of subsequent treatment with a rapidly acting antifilarial drug.

Response: We thank the reviewer for raising these important points. As the risk of SAE with anti-filarial therapy is directly tied to a rapid, massive kill-off of *L. loa* microfilariae in the first 48 hours (as seen with IVM and even more so with diethylcarbamazine), our strategy was to achieve a gradual reduction in microfilarial density (MFD) over the first week. By slowing the decline, we hope to avoid the sudden surge of paralyzed microfilariae that can induce encephalopathic complications, while still lowering counts in patients whose baseline MFD exceeds 30 000 mf/mL. Of course, as the reviewer notes, levamisole's week-long effect may be insufficient in some individuals. In mass treatment, we could imagine measuring again the MFD on D7 after the start of LEV treatment and only proceeding with IVM once the MFD has fallen below the established safety threshold.

Regarding the potential for a drug–drug interaction between LEV and a subsequent course of IVM, in humans, LEV has a rather short half-life (3–4 hours) and is effectively cleared within 24 hours (≈ 6 half-lives) after the last dose. Thus, if IVM is administered at least 24–48 hours after the last day of LEV treatment, virtually all LEV will have been eliminated and hepatically mediated metabolic interactions (both drugs are CYP3A4 substrates) are negligible. From a pharmacodynamic standpoint, LEV and IVM act on different parasite ion-channels (nicotinic acetylcholine vs glutamate-gated chloride), so no synergistic toxicity is expected in the host at the doses used. We added for drug-interaction considerations this paragraph in the Discussion: “Finally, in the context of a mass treatment program, systematic measurement of *L. loa* MFD between LEV and IVM administration would likely not be feasible. Regarding the potential risk of administering the two drugs 24–48 hours apart, we believe this risk is negligible. Indeed, levamisole's plasma half-life is approximately 3–4 hours, yielding essentially complete elimination within 24 hours. If ivermectin is administered at least 24–48 hours after the last levamisole dose, overlap of levamisole exposure is negligible, and CYP3A4 mediated metabolic interactions are unlikely. However, previous studies have reported clinically significant drug–drug interactions when levamisole and ivermectin are co-administered, including increased AUC and C_{max} of ivermectin and reduced AUC and C_{max} of albendazole sulphoxide [34]. Importantly, levamisole and ivermectin exert their effects on distinct parasite ion channels—nicotinic acetylcholine and glutamate-gated chloride channels, respectively—minimizing the likelihood of enhanced host toxicity. However, we therefore consider the risk of meaningful pharmacokinetic or pharmacodynamic interactions in our sequential dosing regimen to be minimal, provided a 24–48 hours wash out is observed.”

Reviewer #3 (Remarks to the Author):

This is a randomized controlled trial evaluating two new regimens of levamisole in comparison to a placebo control for the treatment of individuals with *L. loa* MFD. This is an important trial that is constructed appropriately and the paper is generally well-written. The findings do support the conclusion that both 3 and 5-day regimens of LEV are safe and effective. However, I have many concerns related to the presentation of the results, some of the conclusions drawn, and the methods. Specific comments are noted below, but in particular I do not believe the current presentation of results support the claim that the 5-day LEV protocol is best particularly for individuals with $>30,000$ mf/mL and I do not understand why the authors did not use repeated measures models to evaluate changes in MFD over time.

Abstract

- 1) Several key pieces of information are not included in the abstract, namely a brief mention of statistical methods used and the study's sample size

Response: We have added the requested information in the abstract.

- 2) The following conclusion stated in the abstract does not appear to be supported by the results presented: "A 5-day course of LEV appears to be an option for the individualized management of patients with *L. loa* MFD exceeding 30,000 microfilariae per mL of blood."

Response: We modified the abstract and slightly modified the conclusion.

Results

- 3) It seems that the 1 individual with 0 mf/mL *L. loa* MFD should probably be excluded from all analyses (safety and efficacy)?

Response: As indicated in the Methods and Results sections, as well as in Figure 1, we excluded this individual from the efficacy analysis (both mITT and PP populations). However, they were retained in the safety analysis and for all follow-up assessments (see Figure 1).

- 4) Please also include %s for Sex in Table 1

Response: Done.

- 5) SDs should always be reported alongside means (line 113) and IQRs should always be reported alongside medians (lines 119-124)

Response: Done.

- 6) Depending on which is considered the primary analysis of interest, I would recommend including information on either the mITT or PP analyses in the results section, as opposed to both. Whichever is not included in the main body of the text can be moved to supplemental material.

Response: Our primary analysis was the mITT. We have now added the PP analysis description and results as tables in the supplementary material. In the Results section, we only reported the main estimates for the mITT population and referred the PP details to the supplementary material. In the main text in the Results section, we mentioned: "Finally, PP analysis (Supplementary Information. Tables 2 and 3) showed results consistent with those of the mITT analysis, although the significant difference for comparisons of reduction percentages at 80% was not confirmed."

- 7) As the methods section states that safety is the primary outcome of interest for this paper, the safety results should be reported first, followed by the efficacy results

Response: We agree with the reviewer and changed the Results section accordingly.

- 8) You may consider restructuring tables 2 and 3 to only include the p-value for the overall three-group comparison and then include superscripts to denote the significant pairwise comparisons.

Response: We thank the reviewer for this suggestion. However, we believe it is important to retain the exact figures for other potential readers. We leave it to the editor to decide whether the table should be simplified, for example by replacing the numbers with superscripts.

9) No need to report p-values for non-significant variables (lines 147-150), particularly given that effect estimates are also not reported.

Response: Done.

10) The purpose of figure 3 and why the information shown is critical to the current study is not clear to me

Response: We have decided to include a figure showing the individual trajectories of all participants, stratified both by treatment arm and by baseline MFD. We believe that making these raw data available will be informative; we leave it to the Editor to judge whether this figure adds sufficient value to the final publication.

11) Why did you examine participants with *L. loa* MFD > 30,000? No justification is provided and this subgroup analysis is not described in the methods section

Response: We thank the reviewer for this clarification. Indeed, we did not describe this in the Methods. We decided to include this additional detail to report the proportion of individuals who had an initial *L. loa* MFD > 30 000 mf/mL and who fell below that threshold after treatment. This information may be informative and useful, as individual treatment recommendations often use this cutoff to decide when to start IVM following an albendazole course.¹² We have added the following to the Methods section: “Finally, given the commonly used threshold of 30,000 mf/mL to determine when to initiate IVM treatment after an albendazole course, we reported the proportion of individuals with *L. loa* MFD > 30,000 mf/mL before treatment who fell below this threshold at D5 and D7.”

12) The sentence in lines 186-188 needs to be revised for clarity.

Response: Done. “In other words, a total of 69 adverse events (AEs) were reported by 54 participants, all of which were considered possibly related to treatment, with some individuals experiencing more than one AE.”

13) Does Table 5 show the number of unique AEs of each type or the number of unique subjects reporting each AE type?

Response: In the revised version, Table 3 shows the number of unique AEs of each type, not the number of participants reporting each AE type. We have revised the table footnote for clarity: “Data presents the total number of AEs by symptom type, with percentages representing the proportion of individuals experiencing each symptom in each group. For the summary by organ systems (gastrointestinal, eye, skin, general disorders, and musculoskeletal disorders), percentages represent the proportion of individuals who reported symptoms within each category among the total number of individuals included in each group.”

14) Full results from the logistic regression model predicting AE occurrence (lines 208-216) should be reported in a table

¹² Boussinesq M. Loiasis: new epidemiologic insights and proposed treatment strategy. J Travel Med. 2012 May-Jun;19(3):140-3. doi: 10.1111/j.1708-8305.2012.00605.x. PMID: 22530819.

Response: We thank the reviewer for this comment, we have now added a new table for these results (now Table 4).

Discussion

15) The description of results from Figure 3 (lines 252-260) should be presented first in the Results section and then commented on in the Discussion section

Response: Done.

16) I do not see where the following statement on lines 287-289 is substantiated in the Results: “In this trial, the proportion of participants transitioning from an MFD exceeding 30,000 mf/mL to an MFD below this threshold at D5 or D7 was similar in the LEV-3 and LEV-5 groups”

Response: We changed the text in the penultimate paragraph of the Results section to present the results of the tests that were not significant but whose interpretation should be made with caution given the very small sample sizes: “Among participants with > 30,000 mf/mL at D1, the proportions falling below 30,000 mf/mL by D5 were 28.6% (2 of 7) in PLA, 60.0% (3 of 5) in LEV-3, and 50.0% (2 of 4) in LEV-5 (Fisher’s exact test for LEV-3 vs LEV-5: $P = 0.643$). By D7, these proportions were 33.3% (2 of 6) in the PLA group, 33.3% (1 of 3) in the LEV-3 group, and 50.0% (2 of 4) in the LEV-5 group (Fisher’s exact test for LEV-3 vs LEV-5: $P = 0.629$).” And we have added this part of analysis in the Methods section: “Finally, given the commonly used threshold of 30,000 mf/mL to determine when to initiate IVM treatment after an albendazole course, we report the proportion of individuals with *L. loa* MFD > 30,000 mf/mL before treatment who fell below this threshold at D5 and D7.”

17) Similarly, and in line with my comment on the abstract, I do not see how the results support the authors’ conclusion that the 5-day protocol is best for individuals with > 30,000 mf/mL MFD.

Response: We added the requestion information in the results section: “Post hoc analyses (Table 6) revealed that, beyond the significant effects of the LEV-3 and LEV-5 groups compared to PLA, the LEV-5 group was significantly more effective than the LEV-3 group.” And “Among participants with > 30 000 mf/mL at D1, the proportions falling below 30,000 mf/mL by D5 were 28.6% (2 of 7) in PLA, 60.0% (3 of 5) in LEV-3, and 50.0% (2 of 4) in LEV-5 (Fisher’s exact test for LEV-3 vs LEV-5: $P = 0.643$). By D7, these proportions were 33.3% (2 of 6) in PLA, 33.3% (1 of 3) in LEV-3, and 50.0% (2 of 4) in LEV-5 (Fisher’s exact test for LEV-3 vs LEV-5: $P = 0.629$) (Fig. 5).”

In the discussion, we added: “In this trial, the proportion of participants transitioning from an MFD exceeding 30,000 mf/mL to a MFD below this threshold at D5 or D7 was similar in the LEV-3 and LEV-5 groups; although the small number of individuals with microfilarial densities above 30,000 mf/mL may account for the absence of statistical significance for this outcome. However, based on all of our results including the higher significant effect of LEV-5 vs LEV-3 at D5 and D7 in our mixed model,....”.

However, we slightly adapted the abstract (see new proposition) in the conclusion of the abstract: “A 5-day course of LEV appears to be a safe and relatively effective option for the individualised management of patients with high *L. loa* MFD”.

Methods

18) I assume the formula for calculating reduction rates should include multiplying the proportion by 100?

Response: We have added this information.

19) The purpose of the logistic regression as described in lines 407-410 is unclear. Also, why was *L. loa* MFD categorized in this way instead of using the continuous value or the same categorizations for the efficacy analysis? It is also unclear whether controlling for percentage reduction of MFD at D3 would be appropriate if AEs could have occurred prior to D3?

Response: See response 22 below.

20) Lines 416-417 indicate that both mITT and PP analyses were conducted. Which can be considered the primary analysis?

Response: We have added “primary analysis” in the statistical section after mITT.

21) Were additional individual characteristics (age, sex, treatment) controlled for in the linear regression described in lines 417-420? Based on the results it seems that they were, but this needs to be described in the methods as well.

Response: See response 22.

22) Why wasn't a repeated measures model used to examine changes in MFD over time?

Response: We thank the reviewer for this important point, and we have conducted this analysis. To simplify, we have removed the previous linear model and now present only this one. We have completed the information on the model structure and its covariates, and we have justified our categorization, which is now consistent for both our model for AEs and efficacy, making our results clearer and more readable. Below is the text added to the Statistical section:

“We used a mixed-effects linear model to assess the percentage of pre-treatment *L. loa* MFD retained after treatment. The model included treatment group, time point, and their interaction as fixed effects, with adjustments for age, sex, and baseline MFD category (<20,000, 20,000–29,999, and >30,000 mf/mL). Random intercepts and slopes for time points were specified for each participant to account for within-subject correlations, using an unstructured covariance matrix. The interaction between time points and treatment group was assessed with a likelihood ratio test. Baseline *L. loa* MFD was categorized because the linearity test was not significant, and this categorization was validated against the continuous variable using AIC and BIC. After fitting the model, we estimated our ICC. Finally, post hoc contrasts were performed to compare treatment efficacy between pairs of groups at each time point, assessing the significance of these differences (applying a Wald's test). Finally, given the commonly used threshold of 30,000 mf/mL to determine when to initiate IVM treatment after an albendazole course, we reported the proportion of individuals with *L. loa* MFD > 30,000 mf/mL before treatment who fell below this threshold at D5 and D7.”

And we have added in the result section:

“Our multivariable mixed model over time showed that the interaction between time and treatment groups was significant ($P < 0.001$), and model fit was substantially improved when including a random slope for time (Akaike and Bayesian information criteria – AIC 4906 and BIC 5017) compared to models without a random slope for time (AIC 4955 and BIC 5057). Age and sex were not significantly associated with changes in *L. loa* MFD over time (Supplementary Information. Table 1). In contrast, higher baseline *L. loa* MFD was significantly associated with higher post-treatment values. Post hoc analyses (Table 7) revealed that, beyond the significant effects of the LEV-3 and LEV-5 groups compared to PLA, the LEV-5 group was significantly more effective than the LEV-3 group. Specifically, as illustrated in Fig. 3, individuals in the LEV-5 group showed a 14.9% and 23.8% greater reduction in pre-treatment *L. loa* MFD at D5 and D7, respectively, compared to those in the LEV-3 group.”

“Individual *L. loa* MFD trajectories (Fig. 4) illustrate low inter-individual variability in the LEV-5 group, despite two individuals in the >20,000 mf/mL category who showed a rapid re-increase in MFD after Day 3–5. Overall, these individual trajectories were generally homogeneous, particularly during the first week in the intervention groups. In addition, we estimated an intraclass correlation coefficient (ICC) of 51.3% (95% CI 38.8–63.6) for the full model. However, when the model was restricted to the first week of follow-up, the ICC dropped to 5.7% (95% CI 0.7–33.2). This finding suggests very low inter-individual variability in response to treatment during the first week, with greater variability appearing later.”

REVIEWER COMMENTS

I commend the authors for their careful revision of this manuscript. The majority of my prior concerns have been adequately addressed. I just have a few minor comments remaining.

- The revised sentence on lines 131-133 is still odd. Inherently by nature of the fact that 54 participants reported 69 AEs, there must have been multiple AEs reported by some individuals, so the phrase “with some participants experiencing more than one AE” is extraneous. It might instead be useful to know the variability in how many AEs were reported total.

Response: We thank the reviewer for this insightful comment and agree that the original phrasing was redundant. We have revised the sentence to provide a clearer and more informative description of the distribution of adverse events (AEs) among participants: “A total of 75 adverse events (AE) were reported during the trial. Of these, six were considered unrelated to the intervention, based on chronologic or clinical criteria: one long-standing pruritus, one case of lumbago in the context of chronic back pain, one known gastritis, one pneumonia, one uterine fibroma, and one possible viral rhinitis. The remaining 69 AEs, all considered possibly related to the intervention, were reported by 54 participants. Among these, 41 participants experienced a single AE, 11 experienced two AEs, and 2 participants reported three AEs.”

- In Table 3, reporting the frequency of AEs but the percentage of individuals is very confusing. I would stick with one or the other, either the number of unique individuals reporting that AE category and associated % of individuals or number of unique AEs and % out of total AEs.

Response: Thank you for this suggestion. We acknowledge that presenting both the number of adverse events and the percentage of individuals can be potentially confusing. To improve clarity, we have simplified Table 3 by removing the percentages associated with individual symptoms and retaining only the percentages for organ-level AE categories. However, we believe that both types of information (number of AEs and % of individuals affected) can provide complementary insights, especially in the context of safety monitoring. Therefore, we would prefer to maintain the current structure unless the editor recommends a more standardized format. We are happy to revise accordingly if needed.

- Please explicitly state the how the dependent variable for the linear mixed effects model (“percentage of pre treatment L. loa MFD retained after treatment”) was calculated, as this appears to be different from the reduction rate specified in the “Objectives and outcome measures” section.

Response: We thank the reviewer for this observation. We have clarified in the manuscript how the dependent variable used in the linear mixed-effects model—“percentage of pre-treatment Loa loa microfilarial density (MFD) retained after treatment”—was calculated. Specifically, this percentage was computed as follows:

$$- \quad (\text{MFD at follow-up} / \text{MFD at baseline}) \times 100$$

This approach differs from the “reduction rate” described in the “Objectives and outcome measures” section, which was calculated as:

$$- \quad [(\text{MFD at baseline} - \text{MFD at follow-up}) / \text{MFD at baseline}] \times 100$$

- The acronyms AIC, BIC, and ICC should be spelled out prior to first use.

Done.